

# Distributed search and fusion for wine label image retrieval

Xiaoqing Li[1,2] and Jinwen Ma[2]

[1] School of Statistics, Capital University of Economics and Business, Beijing, China
[2] School of Mathematical Sciences and LMAM, Peking University, Beijing, China

## ABSTRACT

With the popularity of wine culture and the development of artificial intelligence (AI) technology, wine label image retrieval becomes more and more important. Taking an wine label image as an input, the goal of this task is to return the wine information that the user hopes to know, such as the main brand and sub-brand of the wine. The main challenge in wine label image retrieval task is that there are a large number of wine brands with the imbalance of their sample images which strongly affects the training of the retrieval system based on deep learning. To solve this problem, this article adopts a distribted strategy and proposes two distributed retrieval frameworks. It is demonstrated by the experimental results on the large scale wine label dataset and the Oxford flowers dataset that both our proposed distributed retrieval frameworks are effective and even greatly outperform the previous state-of-the-art retrieval models.

## INTRODUCTION

In our daily life, wine is one of the most widely consumed beverages in the world, so it is of great significance and valuable to realize an automatic wine label image retrieval system. In fact, wine label image retrieval can also promote the development of wine e-commerce. Consumers input the wine label images captured by their devices into a shopping website with a wine label image retrieval system, and then search and buy the wines they want. This automatic retrieval system can save customers' shopping time and improve shopping experience. For example, Alibaba's Pailitao (*Zhao et al., 2019*), JD.com (*Li et al., 2018*), Walmart (*Magnani et al., 2019*) can all complete the task of wine label image retrieval. Wine label image retrieval can also help people understand wine culture better. As shown in Fig. 1, when users encounter unfamiliar wines, they can obtain wine-related information through a wine label image retrieval system, such as the manufacturer (main-brand), the type of wine (sub-brand), the productive year and so on. Therefore, it is necessary to build an effective wine label image retrieval system to better serve our production and life. An effective wine label image retrieval system is that when a user inputs a query image, the system can return related wine label images quickly and accurately from the specified wine label dataset. However, it is not easy to build this retrieval system because there are two main challenges in wine label image retrieval task. First, there are a huge number of wine label images, with large numbers of main-brands and sub-brands, which makes the retrieval task more complicated. Second, the numbers of samples from different

Corresponding author
Jinwen Ma, jwma@math.pku.edu.cn

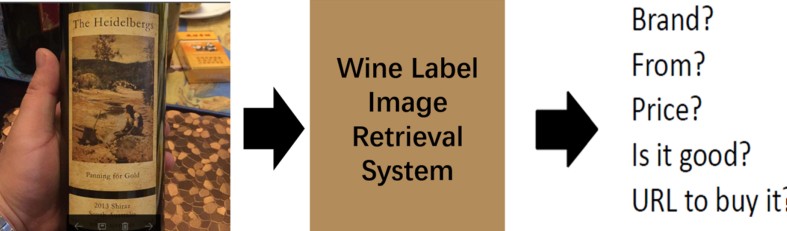

**Figure 1 Wine label image retrieval.**

main-brands varies greatly, and the numbers of samples from different sub-brands are also quite different, even some brands have only one sample. This inter-class imbalance can lead to poor retrieval performance when CNN based methods are used for this task.

So far, some researchers have made some efforts for wine label image retrieval. According to the type of fetures they used, these efforts can be classified into three typical categories: conventional feature based methods, convolutional neural networks (CNN)-based methods and fusion methods. Conventional feature based methods are mainly used in the early wine label image retrieval systems (*Jung et al., 2011*; *Lim et al., 2009*). *Lim et al. (2009)* used an edge-based method to get the wine label region, and then implemented fuzzy c-means clustering (*Bezdek, Ehrlich & Full, 1984*) on its individual wine letters to recognize the text. However, this method has poor generalization and unstable performance, because it works only if the texts on the wine label are in English and it relies heavily on the location accuracy of wine label areas. *Wu, Lee & Kuo (2015)* utilized the Speeded Up Robust Features (SURF) descriptors (*Bay, Tuytelaars & Gool, 2006*), K-D tree (*Zhou et al., 2008*) and K-means method to build a client-sever searching architecture for wine label image retrieval. It performs well on small datasets. When searching from a large wine label image dataset, the searching retrieving system becomes impractical because matching SURF features on large datasets is time-consuming. In recent years, with the increased interest of convolutional neural networks, CNN based image retrieval approaches have become active. These approaches can reduce the semantic gap in image retrieval compared with conventional retrieval methods. Their features are extracted from fully connected layers (*Razavian et al., 2016*) or convolutional layers (*Zheng et al., 2015*), and then employed to match with the methods such as SVM or softmax regression. In addition to extracting features from the whole image, CNN models also extract the features from local regions of the image. At present, most methods use deep metric learning (*Zeng et al., 2020*; *Zheng, Liu & Yin, 2021*) to extract more distinguishing features and achieve good performance. For example, Proxy-Anchor (*Kim et al., 2020*), MPFE (*Cao, Zhu & Lu, 2021*) and HDCL (*Zeng et al., 2021*) are all classic and effective retrieval methods in general image retrieval problems. However, they are not suitable for wine label image retrieval tasks. This is due to the inter-class imbalance in large wine label image datasets. In these wine label image datasets, the numbers of samples from different main-brands are different, and the numbers of samples from different sub-brands are also quite different. Even some brands only have one or two samples. This serious inter-class imbalance makes it impossible to train an effective CNN model for wine label image retrieval. So, the simplex

CNN-based algorithms are not effective for wine label image retrieval in large datasets. To solve above problem, fusion methods based on conventional feature and CNN began to appear. *Li, Yang & Ma (2019)* proposed CNN-SIFT Consecutive Searching and Matching (CSCSM) framework (*Li, Yang & Ma, 2020*) and CNN-SURF Consecutive Filtering and Matching (CSCFM) framework (*Li, Yang & Ma, 2020*) for wine label image retrieval on large-scale wine label image datasets. Both frameworks are two-phase retrieval frameworks, which can not only retrieve the main-brand but also find out the sub-brand about wine. In particular, the CSCSM framework firstly utilizes a deep CNN model to shrink the searching range by recognizing the main-brand in a supervised learning mode, and then applies an improved SIFT descriptor to match the sub-brand about wine. The CSCFM framework improves and extends the study of the CSCSM framework methodologically and theoretically. It utilizes a new version of CNN architecture and an improved SURF matching strategy with modified TF-IDF distance to reduce the computational cost and improve the retrieval performance greatly. Although the above two-phase retrieval frameworks have achieved good retrieval results, the inter-class imbalance about main-brands from wine label image datasets has not been completely resolved in CNN training, which will affect the accuracy of identification for wine main-brands by CNN model. In addition, when the number of wine brands is huge and the inter-class sample sizes is imbalance, the ability of CNN models to learn from datasets is limited, even if the datasets have a lot of training data. Therefore, the CSCFM framework also has certain limitations in large-scale wine label image retrieval. On the other hand, to the best of our knowledge, there is no algorithm that can effectively solve the problem that the inter-class imbalance in large-scale wine label image datasets affects the effect of wine label retrieval. This is also a problem that needs to be solved as soon as possible in the practical application of this field. Therefore, in order to solve the above problems, this article constructs two distributed wine label image retrieval frameworks based on the CSSM framework. They can not only further reduce the impact of inter-class imbalance on CNN model training, but also enable multiple CNN models to learn more fully from the training datasets simultaneously. The retrieval frameworks proposed in this article improve the retrieval accuracy of wine image. In addition, it is also instructive for other fine-grained image retrieval tasks in large-sacle datasets with inter-class imbalance problem.

In machine learning, distributed strategies are usually used to train models (*Campos et al., 2017*; *Lakhan et al., 2021*; *Vlimant & Yin, 2022*), and the data parallel strategy (*Gnip, Vokorokos & Drotár, 2021*; *Wei et al., 2021*) is one of the most commonly used distributed strategies in models training. The data parallel strategy first places multiple copies of the same model on different devices, then allocates different data to each device, and finally merges the results of all devices in a certain way. In the case of large-scale training data, it can improve training efficiency. Our distributed retrieval algorithm is inspired by this, which can perform retrieval tasks quickly and well. On the one hand, we distribute the wine label image data into several parts and allocate them to multiple copies of the same retrieval model for CNN training. On the other hand, the wine label image data is distributed in consideration of the distribution of class sample sizes about wine main-brands, and then the data enhancement and training are performed according to the actual

situation of the data on each device, respectively. In this way, the inter-class imbalance in training data can be further reduced, thereby our distributed retrieval algorithm can improve the recognization accuracy of CNN model and final wine label image retrieval accuracy.

The main contributions of this article are as follows:

- Based on the proposed CSCFM framework, we introduce a distributed strategy and proposed two distributed retrieval frameworks specifically for wine label image retrieval on large-scale datasets.
- Our distributed retrieval frameworks partition data by considering its distribution, which can reduce the impact of inter-class imbalance for CNN training. This data partitioning strategy can improve the accuracy and speed of identifying the main-brand of wine.
- In the distributed retrieval frameworks, we propose several multi-branch result fusion strategies, which improves the ability of model to search wine sub-brands.
- Experiments on the existing large-scale wine label image dataset show that the distributed retrieval frameworks proposed by us can complete the wine label image retrieval task faster and better. Moreover, our frameworks can be effectively generalized and applied to other class-imbalanced fine-grained image retrieval tasks.

## DISTRIBUTED RETRIEVAL FRAMEWORKS

This article introduces a distributed strategy to the CSCFM retrieval framework and proposes two distributed retrieval frameworks specifically for wine label image retrieval on large-scale datasets. We first simply review the CSCFM framework (*Li, Yang & Ma, 2020*). It is a client-server system. In the client site, a user puts a wine label image into the server site as a query image. In the server site, a CNN model is used to segment the wine label area in the query image. Then, CSCFM uses a fine-tuned DPN network model to return some possible main-brands. Next, the query image is matched with all images in possible main-brands by improved SURF matching, respectively. Finally, the most similar images are returned to the client site.

Although the CSCFM retrieval algorithm has achieved good retrieval results, it is not difficult to find that it still has some problems. On the one hand, although the data was augmented before the CNN training, it only impaired the influence of the inter-class imbalance to a certain extent. The problem of imbalanced sample size still exists and has not been solved in essence, which will affect the effective training and parameter learning of classification CNN model. On the other hand, there are a large number of training data, and the number of data categories is too large in large-scale data sets. In addition, the sample size of each category in the dataset is limited and imbalanced, so that the classification CNN model can learn limited knowledge from the dataset and its learning ability is also limited.

In order to solve the above problems, we introduced a distributed strategy based on the CSCMF framework and proposed a distributed retrieval algorithm. The distribution here is

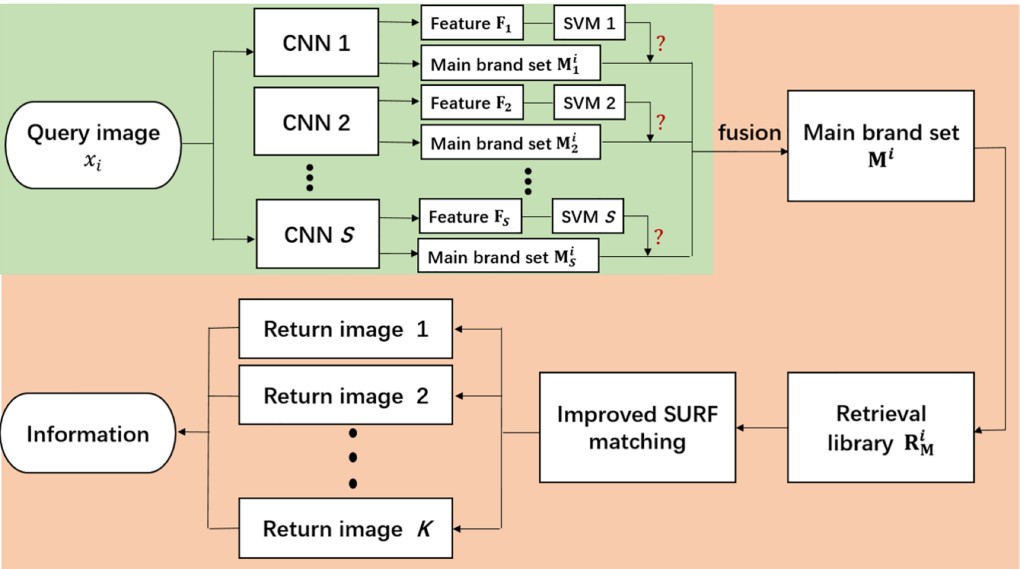

**Figure 2 The overall flow of the distributed retrieval framework based on fusion CNN.** The green section is the data parallel processing in the retrieval framework, and the orange section is the fusion and post-processing of the data parallel processing results.

mainly for data parallelism, that is, in the input stage of the training of distributed retrieval framework, the data should be divided into several parts. We divide the data with the following strategy. First, we ranked the wine main-brands according to their sample sizes in the wine label image dataset (from less to more). Then, based on the above order, the wine main-brands are equally divided into $S$ parts. Finally, all the samples under the main-brands contained in each part constitute the dataset of this part, and then the training dataset of this part is input to the model corresponding to this part for model training. For the dataset of each part obtained in this way, the sample size of different classes are relatively balanced, and the number of main brands of wine is appropriate. Therefore, each classification CNN model can carry out effective training and parameter learning, so as to comprehensively and fully learn the feature information in the image from the dataset.

After the query image is input into the distributed retrieval algorithm in parallel, the algorithm needs to fuse the processing results of different branches to get the final retrieval results. According to the different fusion strategies of different branch results in the retrieval framework, we propose two distributed retrieval frameworks: the distributed retrieval framework based on fusion CNN (DCSCFM1) and the distributed retrieval framework based on fusion SURF matching (DCSCFM2). Next, We first introduce the distributed retrieval framework based on fusion CNN. Its overall flow is shown in Fig. 2. The green part is the data parallel processing part in the retrieval framework, and the orange part is the fusion and post-processing part of the data parallel processing result. Its training stage is divided into two steps. The first step is to train the classification CNN models on each branch. In this step, the algorithm first divides the dataset into $S$ subsets, and then augments the data according to the actual situation of the training dataset of each

subset. The data augment strategies include adding Gaussian blur to the image, changing contrast, sharpness, saturation, brightness, and so on. Finally, the *S* augmented training subsets are input into the *S* classification CNN models for training. It is worth noting that the initial parameters of the *S* classification CNN models here are all the same, and they are all copies of the model obtained after pre-training on the ImageNet dataset (*Krizhevsky, Sutskever & Hinton, 2017*). After training on distributed data, we obtained *S* different trained CNN models. The second step in the training phase is to train the SVM model (*Hearst et al., 1998*) on each branch. The SVM model in each branch is used to determine whether the class of the query image belongs to the branch. For each branch, we first input all training images (the total training dataset without data partition) into the trained CNN model on the branch respectively to extract their CNN features. Since the dimension of output vector in the last full-connection layer of the CNN model on each branch is large and not necessarily the same, we generally take the output of the input image on the second-to-last full-connection layer of the CNN model as the CNN feature of the image. Then, the CNN features of all the training images are input into the SVM model on this branch for training. Here, the SVM model is a binary classifier. If the category of the image represented by CNN features just belongs to the range of the main brands of wine classified that is divided into this branch in the first step of the training phase, it will be judged as belonging to this branch. Otherwise, it is determined not to belong to the branch. After the same training for all the branches, we obtained *S* different trained SVM models. In the test phase, after the query image $x_i$ (where $1 \leq i \leq N$, *N* is the number of images in the query dataset) is input into the distributed retrieval framework, it first needs to go through the parallel processing of *S* branches. For the purpose of clarity, we assume that the query image enters the *s* branch in the retrieval framework, where $s = 1, \cdots, S$. Then the process on this branch is as follows: it first goes through the classification CNN model that has been trained on this branch:

$$O_s^i = f_s(x_i) \tag{1}$$

where, $f_s(\cdot)$ represents the processing operation of the CNN classifier on the *s* branch. The output $O_s^i$ contains two parts, one is the CNN feature $F_s^i$ of the query image $x_i$, and the other output is the main brand range $M_s^i$ to which the wine in the query image $x_i$ may belong:

$$O_s^i = (F_s^i, M_s^i) \tag{2}$$

Next, we input CNN features $F_s^i$ into the trained SVM model on this branch to predict whether the query image belongs to this branch:

$$(y_s^i, p_s^i) = g_s(F_s^i) \tag{3}$$

Our SVM model here is a two-classifier that can output the prediction result and its posterior probability at the same time. $g_s()$ represents the processing operation of the SVM model on the *s* branch. Its output contains two parts, one part is the prediction category $y_s^i$ for the input feature $F_s^i$. If $y_s^i = 1$, it indicates that the category of $x_i$ belongs to this branch;

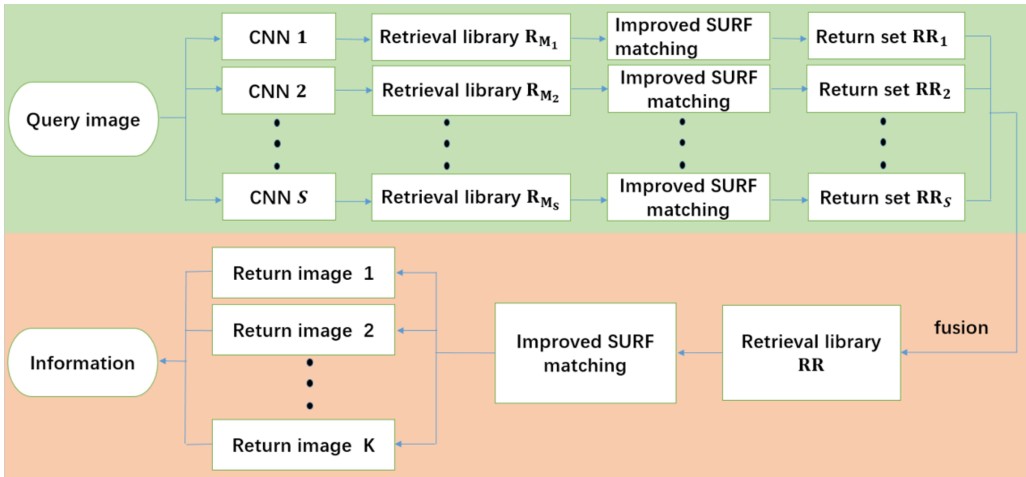

**Figure 3** The overall flow of the distributed retrieval framework based on fusion surf matching. The green section is the data parallel processing in the retrieval framework, and the orange section is the fusion and post-processing of the data parallel processing results.

the other part is the posterior probability $p_s^i$ for the predicted category. In order to prevent the wrong filtering out of the right wine main brand, we also develop the following strategy:

$$q_s^i = \begin{cases} 1, \ y_s^i = 1 \ or \ \{y_s^i = -1, \ p_s^i < \mu\} \\ 0, \ otherwise \end{cases} \tag{4}$$

where, $\mu$ is the threshold we set in advance. When the SVM model judges that the input does not belong to the branch but the posterior probability of the classification is lower than the threshold $\mu$, we still merge the result of the branch into the post-processing part. $q_s^i$ is used to guide the fusion of branch results. If $q_s^i = 1$, it indicates fusion, that is, the above main brand range $M_s^i$ can be merged into the subsequent general main brand range $M^i$: $M^i \leftarrow M^i \cup M_s^i$. If $q_s^i = -1$, it indicates no fusion, that is, the range of main brands of wine predicted on this branch will be ignored. After parallel processing of $S$ branches, the query image arrives the post-processing part. Firstly, all samples under the main brands set $M^i$ constitute the retrieval database $R_M^i$ of the post-processing part. Then, on this database, improved SURF feature matching is implemented on the query images to return most similar image, so as to obtain information about the main-brand and sub-brand of the wine in the query image.

The distributed retrieval framework based on fusion surf matching has different fusion strategy, and its overall flow is shown in Fig. 3. The data processing in the training phase of the distributed retrieval framework based on fusion surf matching is the same as the distributed retrieval framework based on fusion CNN. After the data is divided into $S$ parts, the data in each part is augmented according to their respective data conditions, and then the augmented data is input to $S$ CNN models for training, finally $S$ different trained CNN models are obtained. In the test phase, after the query image is input into the distributed

framework, it is processed in parallel by $S$ trained CNN models. Assuming that the output about the wine main brand set after CNN model on the $s$ branch is $\mathbf{M_s}$, where $s = 1, \cdots, S$. and all the data under the wine main brands in this set $\mathbf{M_s}$ constitute the retrieval dataset $\mathbf{R_{M_s}}$. Next, improved SURF matching is performed between the query image and each image from $\mathbf{R_{M_s}}$ to obtain the candidate set $\mathbf{RR_s}$. The candidate set results obtained from these $S$ branches are fused to obtain a total candidate set $\mathbf{RR} = \mathbf{RR_1} \cup \cdots \cup \mathbf{RR_s} \cup \cdots \cup \mathbf{RR_S}$. Finally, the improved SURF matching is employed again in this RR dataset with the test image, and the most matching images are returned, so as to obtain the desired information about main brand and sub-brand.

## EXPERIMENTS

### Dataset and evaluation metrics

The dataset used in experiments is the large-scale wine label image retrieval dataset provided by Ruixun Science and Technology (Beijing) Limited Company in China. It has 547,857 wine images labeled with 17,328 main-brands and 260,579 sub-brands. These images are manually taken by buyers with mobile phones or other electronic camera devices for wine bottles at anytime and anywhere. They are all formatted into RGB and their sizes are resized into 500 * 375. Each image is labeled with a main-brand and a sub-brand. To understand the dataset, we show some image samples of the dataset in Fig. 4, and the data distribution about the main-brands and sub-brands on this dataset is given in Table 1. From the Table 1, we can realize that there is a serious problem in the dataset. The sample sizes of different main-brands and sub-brands are highly uneven, which is also one of the main problems solved by our distributed retrieval framework. Before the experiments, we randomly divided this data set into two parts, 80% of it as training set and 20% of it as test set.

In the experiments, we take each image in test set as the query image, and then rank all the retrieved images returned by the system according to their similarity to the query image. All experiments use Average Precision (AP) (*Revaud et al., 2019*) to evaluate the retrieval performance of each retrieval algorithm.

### Implementation details

In this subsection, we will introduce some implementation details in experiments, including: data partitioning strategy, data preprocessing and experimental configuration.

Before training the model, we first need to divide the dataset into $S$ parts according to a data partitioning strategy. First, we ranked the wine main-brands according to their sample sizes in the wine label image dataset (from less to more). Then, based on the above order, the wine main-brands are equally divided into $S$ parts. Finally, all the samples under the main-brands contained in each part constitute the dataset of this part, and then the training dataset of this part is input to the model corresponding to this part for model training. If there is no special designation, we will set $S$ to four. After the training set is partitioned, we obtain four training subsets, and their information is shown in Table 2. For example, the sample sizes of main-brands in sub-datatset 1 are only from 11 to 14. The total number of images and main-brands in subset 1 are 51,468 and 5,136, respectively. In

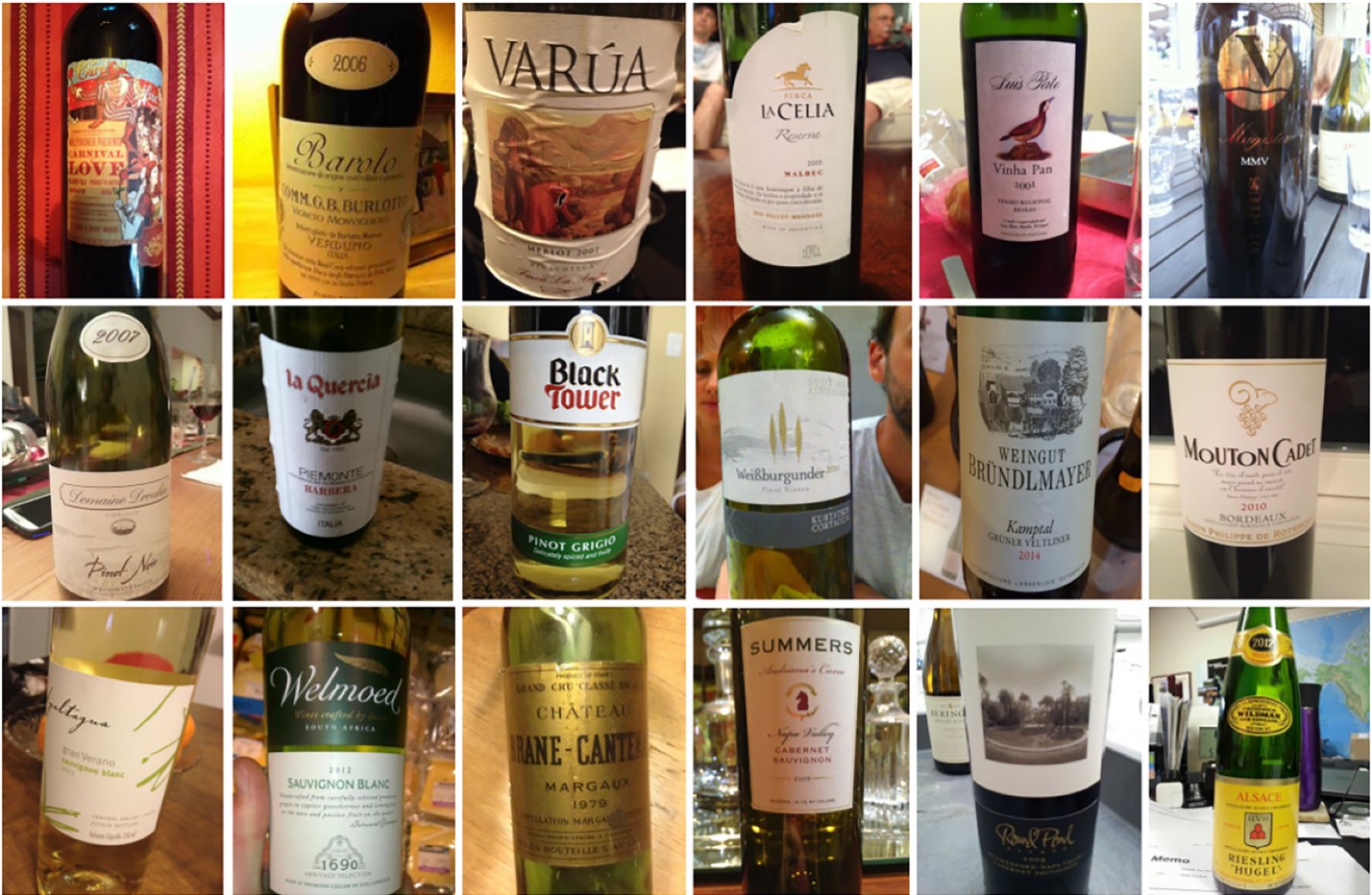

**Figure 4** Some image instances of the large-scale wine label dataset.

**Table 1 The numbers of samples for the main-brands and sub-brands in the dataset.**

| Number of images | 11~20 | 21~30 | 31~50 | 51~100 | 101~1,371 |
|---|---|---|---|---|---|
| Number of main-brands | 8,974 | 3,359 | 2,811 | 1,473 | 711 |
| Number of images | 1 | 2 | 3 | 4~10 | 11~371 |
| Number of sub-brands | 129,932 | 71,257 | 24,993 | 32,802 | 1,595 |

this way, the sample sizes of different main-brands in each subset are more balance, and the number of labels in each sub dataset is more appropriate, which is more benefit to model training. Therefore, each classification CNN model can carry out effective training and parameter learning, so as to comprehensively and fully learn the feature information in the image from the dataset.

After the dataset is partitioned, we firstly perform the data augmentation including adding Gaussian blur (*Hummel, Kimia & Zucker, 1987*), changing contrast, sharpness, saturation, brightness, and tilt for each subset. Because there are many interference factors

**Table 2 The relevant information of four subsets.**

|  | Subset1 | Subset2 | Subset3 | Subset4 |
|---|---|---|---|---|
| NI[a] | 11~14 | 15~20 | 21~42 | 43~1,371 |
| TNM[b] | 5,136 | 4,838 | 4,324 | 4,030 |
| TNI[c] | 51,468 | 79,873 | 157,576 | 259,940 |

Notes:
[a] The number of images under each main brand.
[b] Total number of main-brands in each subset.
[c] Total number of images in each subset.

in datasets, such as background, light changes, local highlight, marginal highlight, image rotation and so on, above data augmentation process can reduce the impact of these interference factors in model training. After that, in order to reduce the interference from the background in model training, we employ a fully convolutional network (FCN) (*Long, Shelhamer & Darrell, 2015*) pretrained by ImageNet to locate the accurate wine label regions, and then use mask images getted by FCN to get accurate wine label images, finally resize these images to $224 * 224$. After the above preprocessing, the data of each subset can be input into CNN model for training. During training, we employ dual path networks (DPN) (*Chen et al., 2017*) as the backbone of CNN model in our distributed retrieval framework. For network optimization, synchronized SGD is used as the optimizer with a momentum of 0.9, weight decay of $10^{-4}$, and batch size of 80. The initial learning rate of SGD is 0.01. In addition, the dropout (*Srivastava et al., 2014*) of 0.5 is used before the final classifier layer. All models are trained for 500 epochs in total. Without otherwise stated, all the experiments are performed on a Linux server with two GPU graphics cards, which is NVIDIA GeForce GTX 1080.

## Experiments on the large-scale wine label image dataset

In this subsection, we compare two distributed retrieval frameworks with several retrieval algorithms which performed well on large-scale wine label image datasets, to show the effectiveness of our proposed distributed retrieval framework. In experiment, we mainly compare the following four algorithms:

- CNN-SIFT Consecutive Searching and Matching (CSCSM) framework. In this framework, two parameters $\delta$ and $M_2^i$ are set to 95% and five respectively, and the backbone network of the classification CNN model is ResNeXt-50.
- CNN-SURF Consecutive Filtering and Matching (CSCFM) framework. In this framework, two parameters $\delta$ and $M_2^i$ are set to 95% and five respectively, and the backbone network of the classification CNN model is DPN-92.
- Object-level Representation (OR) method (*Sun et al., 2015*). It combines CNN features and SIFT features, and encodes them with a Product Quantization (PQ) scheme for image retrieval.
- The distributed retrieval framework based on fusion CNN (DCSCFM1). In this framework, three parameters $S$, $\delta$ and $M_2^i$ are set to four, 95% and three respectively, and the backbone network of the classification CNN model is ResNeXt-50.

**Table 3 The experiment results on the large-scale wine label image dataset. The best performances are marked in bold.**

| Methods | Backbone | MAP[a] | SAP[b] | Time[c] (s) |
|---|---|---|---|---|
| CSCSM | ResNeXt-50 | 91.07 | 78.40 | 9.5657 |
| CSCFM | DPN-92 | 92.23 | 82.32 | 2.8956 |
| OR | ResNeXt-50 | 90.67 | 76.98 | 2.2015 |
| DCSCFM1 | ResNeXt-50 | **93.51** | **84.03** | **1.3596** |
| DCSCFM2 | ResNeXt-50 | 93.16 | 83.36 | 2.2976 |

Notes:
[a] The average retrieval accuracy for wine main-brands.
[b] The average retrieval accuracy for wine sub-brands.
[c] The average retrieval time for each image.

- The distributed retrieval framework based on fusion SURF matching (DCSCFM2). In this framework, three parameters $S$, $\delta$ and $M_2^i$ are set to four, 95% and five respectively, and the backbone network of the classification CNN model is ResNeXt-50.

It is worth noting that the classification CNN models in the above algorithms are all pre-trained on the ImageNet dataset. Their configurations and experiment results are shown in Table 3.

As we can see from the experiment, the OR algorithm combines CNN and SIFT features. However, in the CNN Finetune stage, the intra-class imbalance problem in the large-scale wine label dataset has not been solved. Moreover, there are small inter-class variances and large intra-class variances in wine label dataset. So, the performance of OR algorithm unsatisfactory in the large-scale wine label retrieval task. Although the backbones of DCSCFM1 and DCSCFM2 are ResNeXt-50 which is lighter and simpler than DPN-92, the two frameworks both are better than CSCSM and CSCFM in terms of retrieval accuracy and time. In particular, compared with CSCSM and CSCFM, the average precision for the wine main brand of DCSCFM1 is increased by 2.44% and 1.28%, the average precision for the wine sub-brands is increased by 5.63% and 1.71%, and retrieval time is shorten by 8.2061 and 1.5360 s respectively; the average precision for the wine main brand of DCSCFM2 is increased by 2.09% and 0.93%, the average precision for the wine sub-brands is increased by 4.96% and 1.04%, and retrieval time is shorten by 7.2721 and 0.6020s, respectively. The above results show that the distributed retrieval frameworks proposed in this article are effective for wine label image retrieval task on large-scale datasets. Meanwhile, They also show that distributed strategies can further solve the problem of uneven distribution of sample sizes in large-scale datasets. It can improve the training quality and training efficiency of the CNN model, so that it can learn the image features better and faster. In addition, the experimental results of DCSCFM1 and DCSCFM2 are different. When the number of branches is four, DCSCFM1 retrieval framework can complete the retrieval task better and faster than DCSCFM2 retrieval framework, which is caused by the different fusion strategies in frameworks. In the experiment, for each wine label image, the classification CNN model of each branch in DCSCFM1 returns three possible wine main brands, but not all the main brand sets

**Table 4 The relevant information of two subsets.**

|  | Subset 1 | Subset 2 |
|---|---|---|
| NI[a] | 11~20 | 21~1,371 |
| TNM[b] | 8,974 | 8,354 |
| TNI[c] | 130,341 | 417,516 |

Notes:
[a] The number of images under each main brand.
[b] Total number of main-brands in each subset.
[c] Total number of images in each subset.

obtained by each branch can be integrated into the subsequent retrieval database. It needs to be guided by the SVM model on the branch. If the SVM model predicts that the query image belongs to the branch, then the main brand set output by the branch has opportunity to be integrated into the subsequent retrieval database. That is to say, after the filter process by the previous branches of DCSCFM1, the search library in the post-processing part will be a high-quality reduced search library. There are not too many irrelevant images in the search library, so in the post-processing part, the improved SURF feature matching can be performed quickly and well. Therefore, DCSCFM1 obtains the highest average retrieval accuracy for the wine main brands and sub-brands, and the average retrieval time is also the shortest. As for DCSCFM2, in each branch, the query image first is input into the CNN model to get five possible wine main brands. Next, the first improved SURF feature matching in the search database composed of all samples belonging to the five wine main brands mentioned above is performed in to obtain the three most similar images. After performing the same operation in parallel in the four branches, the algorithm fuses the matching results of the four branches to form the retrieval candidate library again. In this candidate library, the algorithm performs the second improved SURF feature matching operation, and finally returns the retrieval result. The SVM models in DCSCFM1 are trained in advance, so its prediction guidance process is not time-consuming. DCSCFM2 needs to carry out two improved SURF feature matching processing, so it takes longer time. However, its retrieval is gradually accurate from coarse to fine, so the retrieval accuracy of DCSCFM2 for the main brand and sub-brand will also be improved.

## Ablation studies

In this section, we conducted two ablation studies to evaluate the influence of the number of branches in two retrieval frameworks and the effect of the SVM model in DCSCFM1 on the retrieval effect, respectively. Next, we first introduce the ablation experiment that evaluates the influence of the number of branches in the distributed retrieval frameworks on the retrieval effect. In this experiment, we set the number of branches in DCSCFM1 and DCSCFM2 as 2, 4, 8, 16 respectively. Tables 4, 2 and 5 show the related data information when dataset is divided into 2, 4, 8 parts, respectively. In addition, for each retrieved image, the number of possible main brands returned by each classified CNN model branch in DCSCFM1 and DCSCFM2 is set as three and five, respectively. In the subsequent improved SURF matching process of DCSCFM2, each branch returned three most similar

**Table 5  The relevant information of eight subsets.**

|  | Subset 1 | Subset 2 | Subset 3 | Subset 4 |
|---|---|---|---|---|
| NI[a] | 11 | 12 | 13~14 | 15~20 |
| TNM[b] | 2,764 | 2,172 | 2,124 | 2,914 |
| TNI[c] | 30,404 | 26,064 | 28,371 | 45,142 |
|  | Subset 5 | Subset 6 | Subset 7 | Subset 8 |
| NI[a] | 21~27 | 28~48 | 49~66 | 67~1371 |
| TNM[b] | 1,869 | 2,455 | 2,037 | 1,993 |
| TNI[c] | 49,209 | 108,367 | 114,156 | 145,784 |

**Notes:**
[a] The number of images under each main brand.
[b] Total number of main-brands in each subset.
[c] Total number of images in each subset.

**Table 6  Ablations of the number of branches on the large-scale wine label image dataset. The best performances are marked in bold.**

| Method | S[a] | MAP[b] | SAP[c] | Time[d] (s) |
|---|---|---|---|---|
| D-CSCFM1 | 2 | 92.81 | 82.54 | **1.1132** |
|  | 4 | **93.51** | **84.03** | 1.3596 |
|  | 8 | 93.08 | 83.32 | 2.3160 |
|  | 16 | 92.05 | 82.41 | 3.5632 |
| D-CSCFM2 | 2 | 92.33 | 81.81 | 1.9087 |
|  | 4 | 93.16 | 83.36 | 2.2976 |
|  | 8 | 93.23 | 83.21 | 2.4621 |
|  | 16 | 92.49 | 82.56 | 2.7456 |

**Notes:**
[a] The number of branches.
[b] The average retrieval accuracy of the main-brands.
[c] The average retrieval accuracy of the sub-brands.
[d] The average time for each image.

wine label images. Experimental results on the large wine label dataset are shown in Table 6.

In general, as the number of branches increases, the average retrieval accuracy of DCSCFM1 and DCSCFM2 for the main-brands and sub-brands both present a trend of first increasing and then decreasing. As shown in Tables 4, 2 and 5, as the number of branches gradually increases, the imbalance of the brand sample size in each subset gradually decreases. For example, when S = 4, the sample size of each main-brand in subset1 is 11 14. In this way, it enables the CNN model in each branch to learn the features of the image in each subset better. Therefore, the retrieval accuracy of wine labels will naturally rise. However, when the number of branches in the framework becomes very large, the total sample size of each subset will decrease, which will cause overfit CNN model. Therefore, wine label recognition accuracy decreased when S is bigger than four. For DCSCRM1, as the number of branches increases, the search time increases in proportion to the number of branches. This is because with the increase of the number of branches, the SVM model on each branch of DCSCFM1 has limited capacity. In order to

**Table 7 Ablations of SVM model in DCSCFM1. The best performances are marked in bold.**

| Method | S[a] | MAP[b] | SAP[c] | Time[d] (s) |
|---|---|---|---|---|
| DCSCFM1(-SVM) | 2 | 92.01 | 81.32 | 1.6022 |
| | 4 | 93.16 | 83.42 | 2.0134 |
| | 8 | 92.73 | 82.97 | 5.9807 |
| | 16 | 91.67 | 81.96 | 11.6750 |
| DCSCFM1 | 2 | 92.81 | 82.54 | **1.1132** |
| | 4 | **93.51** | **84.03** | 1.3596 |
| | 8 | 93.08 | 83.32 | 2.3160 |
| | 16 | 92.05 | 82.41 | 3.5632 |

**Notes:**
[a] The number of branches.
[b] The average retrieval accuracy of the main-brands.
[c] The average retrieval accuracy of the sub-brands.
[d] The average time for each image.

prevent the SVM model from filtering out some useful branch results by mistake, we set a very small threshold to make the SVM model retain branch results with high possibility as much as possible. So, with the increase of $S$, the subsequent retrieval library will become very big, this will lead to improved SURF feature matching process takes more time, the interference of too much irrelevant images at the same time also can affect the effect of matching. For DCSCRM2, the retrieval time usually remains relatively stable as the number of branches increases. This is because in DCSCFM2, there are two SURF matching operations. The first SURF matching is processed in parallel on each branch. Therefore, even if $S$ becomes larger, the retrieval library is always composed of images under five main-brands. The second SURF matching retrieval library is composed of 3S images. Since $S$ is 16 at most, SURF matching time is negligible. In a word, D-CSCFM1 and D-CSCFM 2 have their own advantages, and the number of branches can affect the retrieval effect.

In order to explore the impact of the SVM model in D-CSCFM1 on the retrieval effect, we conducted the following ablation experiments. In the experiment, we recorded the version that did not add the guidance of the SVM model in DCSCFM1 as: DCSCFM1 (-SVM). In this version, the SVM model does not work, which means that the results of all branches will be merged into the retrieval library in the post-processing part. We carried out comparative experiments for DCSCFM1(-SVM) and DCSCFM1 under the condition that the number of branches in the algorithm framework was set to 2, 4, 8 and 16, respectively. The specific experimental results are shown in Table 7.

As can be seen from Table 7, the performance of DCSCFM1 is generally better than DCSCFM1(-SVM) on average retrieval accuracy for wine main brand, the average retrieval accuracy for wine sub-brands and average retrieval time. Especially when the number of branches is set to two, the DCSCFM1 algorithm improves 0.8% 1.22% and saves 0.4890 s compared with the D-CSCFM1(-SVM) algorithm in the above three indexes, respectively, which indicates that the SVM model is effective in guiding branch result fusion. When the number of branches in the framework is set to 16, the DCSCFM1 algorithm still performs better than the DCSCFM1(-SVM) algorithm, but the improvement is smaller, which is also

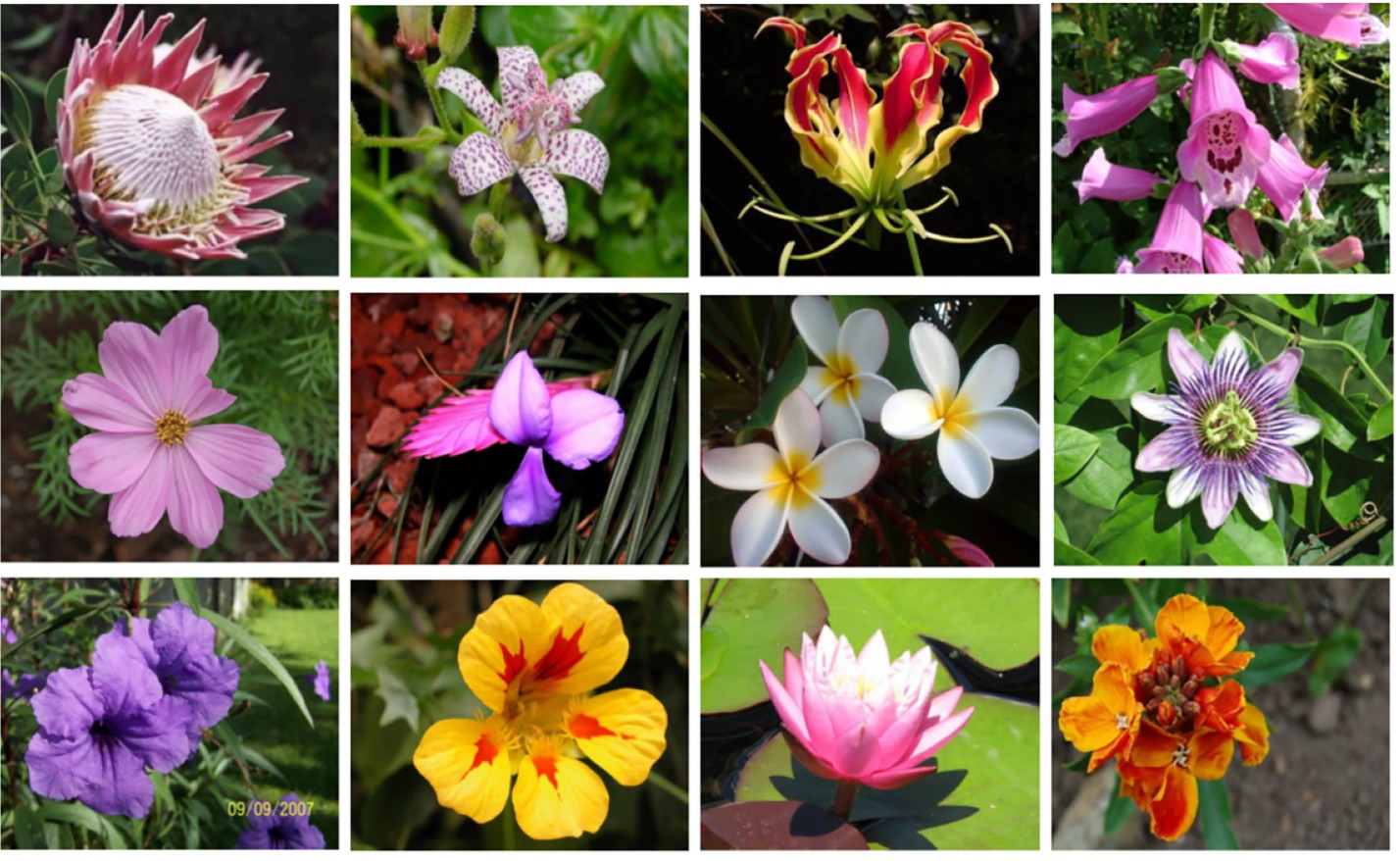

**Figure 5 Typical images of the Oxford flowers dataset.**

related to the reason we analyzed in the last experiment. When the number of branches is too large, the sample size of the category for each SVM model is imbalanced, which makes it difficult for SVM model to be trained effectively. Therefore, the SVM model trained has limited guiding effect on the fusion of branch results, which is also an issue that we should consider to improve in the future.

## Comparison with state-of-the-arts on Oxford flowers dataset

We also evaluate our distributed retrieval frameworks on a public benchmark, Oxford 102 Flowers (*Nilsback & Zisserman, 2008*). The Oxford flowers dataset contains 8,189 images, consisting of 102 flower categories. from the United Kingdom. Each class consists of between 40 and 258 images. This dataset is divided into a training set of 3,680 images and a test set of 4,509 images. The images have large scale, pose, and light variations. The experimental evaluation metric is classic Top-k mAP (where the values of k are one and five), which is widely used for evaluating the retrieval accuracy. Some typical images of the dataset are shown in Fig. 5.

In Table 8, we divide the methods used for comparison into three groups: (1) Three classic methods in general image retrieval task, SPoC (*Babenko & Lempitsky, 2015*), CroW

**Table 8 The experiment results on Oxford flowers dataset. The best performances are marked in bold.**

| Method | Top-1 mAP | Top-5 mAP |
|---|---|---|
| CroW | 73.67 | 76.16 |
| SPoC | 71.36 | 74.55 |
| R-MAC | 71.98 | 74.82 |
| SCDA | 75.13 | 77.70 |
| SGeM | 76.11 | 78.20 |
| DS | 76.09 | 78.15 |
| DCSCFM1 | **77.21** | **79.02** |
| DCSCFM2 | 76.88 | 78.49 |

**Note:**
The numbers of branches for the DCSCFM1 and DCSCFM2 are set to 4.

(*Kalantidis, Mellina & Osindero, 2016*) and R-MAC (*Tolias, Sicre & Jégou, 2015*). (2) Three state-of-the-art methods for fine-grained image retrieval, SCDA (*Wei et al., 2017*), SGeM (*Wang et al., 2018*) and DS (*Lin et al., 2021*). (3) Our DCSCFM1 and DCSCFM1. Note that we do not include methods that using training data or additional annotation. Table 8 gives the experimental results of above methods on the Oxford flowers dataset.

As shown in Table 8, the results of SPoC, CroW and R-MAC are not satisfactory. Although they are classic methods in general image retrieval task, their encoded features lack discriminativeness in fine-grained image retrieval, such as the Oxford flowers dataset. So, compared with the first group, the performance of the second set of methods has greatly improved. DCSCFM1 and DCSCFM1 perform better than SCDA, SGeM and DS. This is because there is a class imbalance problem in the Oxford flowers dataset, which affects the training of retrieval systems. Both DCSCFM1 and DCSCFM1 adopt distributed stragies to solve above problem. Therefore, they can greatly outperform pervious state-of-the-art works. This also shows that our proposed DCSCFM1 framework and DCSCFM1 framework have the ability to be effectively generalized and applied to other class-imbalanced fine-grained image retrieval tasks.

## CONCLUSION

In order to solve the problem of inter-class imbalance in large-scale wine label datasets, this article adopts a distributed strategy for the CSCFM algorithm, and proposes two retrieval frameworks: DCSCFM1 and DCSCFM2. Both frameworks consist of two parts: branch training and fusion post-processing. Branch training can solve the impact caused by inter-class imbalance, and at the same time, and can also filter out some irrelevant data to reduce the retrieval database. Post-fusion processing fuses the results of the previous branches, and then further refines them to obtain the final results. This part can further improve retrieve the results. These two parts complement each other, so that the retrieval results are gradually accurate from coarse to fine. The experiments on the large scale wine label dataset and the Oxford Flowers dataset demonstrate that our proposed two distributed retrieval frameworks, DCSCFM1 and DCSCFM2, are both effective and even greatly outperform the previous state-of-the-art retrieval models. This further shows that our

distributed retrieval frameworks can effectively solve not only large-scale wine label retrieval problems, but also other class-imbalanced fine-grained image retrieval tasks.

In the future, we will continue to expand the application scope of our distributed retrieval frameworks, such as fashion image retrieval, plant image retrieval, bird image retrieval and so on.

### Funding
This work was supported by the Natural Science Foundation of China (No. 62071171) and the Capital University of Economics and Business (No. XRZ2022065). The funders had no role in study design, data collection and analysis, decision to publish, or preparation of the manuscript.

### Grant Disclosures
The following grant information was disclosed by the authors:
Natural Science Foundation of China: 62071171.
Capital University of Economics and Business: XRZ2022065.

### Competing Interests
The authors declare that they have no competing interests.

### Author Contributions
- Xiaoqing Li conceived and designed the experiments, performed the experiments, analyzed the data, performed the computation work, prepared figures and/or tables, authored or reviewed drafts of the article, and approved the final draft.
- Jinwen Ma conceived and designed the experiments, analyzed the data, authored or reviewed drafts of the article, and approved the final draft.

### Data Availability
The codes are available in the Supplemental Files. The data is available at figshare: Li, Xiaoqing (2022): wine label dataset and Oxford 102 Flowers dataset. figshare. Figure. https://doi.org/10.6084/m9.figshare.19786477.v1.

### Supplemental Information
Supplemental information for this article can be found online at http://dx.doi.org/10.7717/peerj-cs.1116#supplemental-information.

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
