# Peer review of "Distributed search and fusion for wine label image retrieval"

_PeerJ Computer Science, doi:10.7717/peerj-cs.1116_

## Round 0.1 · original submission · Minor Revisions

I am happy to inform you that I received the review comments from all reviewers. Based on the review comments I am advising you to revise the manuscript as per reviewer comments and resubmit.

·

Basic reporting

It is a great pleasure to read this paper. From grammar to wording, from literature review to design of experiment, this paper is top tier quality and deserves to be seen by the image recognition community at large.

Having worked in the tech industry for a few years, I cannot count the times when the training data is so small while the number of parameters to estimate is huge. The authors successfully identified this issue in the wine label recognition task. The numerous number of main brands and sub-brands for recognition makes any decent CNN based deep learning models practically impossible to train. The over-fitting issues and the biased sampling issues all render badly designed CNN models perform subpar with testing data.

Another issue challenging both the research community and the industry is the data imbalance in classification category of machine learning models, which this label recognition task falls into. Sorting and parting data into stratified group based on sample size a smart solution during the pre-processing stage in training. The use of SVM and fusion in the paper to avoid over-filtering and biased prediction adds another level of re-assurance for the training data to be less imbalanced.

One more thing worth mentioning is the thoughtful arrangement of the off-line pre-trained SVM in the paper. In both NLP and image recognition fields, I have seen a lot of models with great accuracy metrics, but a common issue is that an easy off-line training is sometimes absent, which severely intervenes the productionization of the said models in reality. The issue is particularly common with deep learning models, such as support vector machines in the paper. I am glad to see that the authors make sure that the threshold $\mu$ for filtering is generated from the pre-trained SVM and not time-consuming.

Experimental design

2 major methods are proposed in the paper, and both get evaluated well. One method is based on CSCFM framework. The other one is distributed retrieval framework based on fusion surf matching. CSCFM is more of a conventional framework, while the latter method introduces noise (sharpness, saturation, ect.) to training data in each branch.

The training and testing data for both methods look carefully and reasonably collected. The sample size is large enough (~548K samples) to assert statistical power (~17K main categories). The 80/20 split for training and testing is widely accepted in statistical analysis. The comparison using Oxford flower data provides a good parallel comparison to other previously existing methods, including SCDA even.

Validity of the findings

Throughout the paper, the findings are well backed by the statistical analysis. The tables are well illustrated and captioned. The numbers are consistent with the authors’ conclusion. Overall, the findings are valid and statistical sounds

·

Basic reporting

no comment

Experimental design

no comment

Validity of the findings

no comment

Additional comments

experimental design : It is well-structured but, according to the reviewer, it could be more implemented via grounded it on real experience

validity of the findings: the validity is robust with the provided statistical analysis (which should be implemented)

The English language should be improved to ensure that an international audience can clearly understand your text.

- Your introduction needs more detail. I suggest that you improve the descriptions to provide more justification for your study (specifically, you should expand upon the knowledge gap being filled).
- I commend the authors for their extensive data set If there is a weakness, it is in the statistical analysis which should be improved upon before Acceptance.

Reviewer 3 ·

Basic reporting

1. Contribution is very clearly written
2. Proposed method clearly stated
3. Satisfactory results part
4. Some more comparison of wine text label detection is needed with the methods used by previous researchers.
5. Future scope should be clearly explained in detail.

Experimental design

1. Contribution is very clearly written
2. Proposed method clearly stated
3. Satisfactory results part
4. Some more comparison of wine text label detection is needed with the methods used by previous researchers.
5. Future scope should be clearly explained in detail.

Validity of the findings

1. Contribution is very clearly written
2. Proposed method clearly stated
3. Satisfactory results part
4. Some more comparison of wine text label detection is needed with the methods used by previous researchers.
5. Future scope should be clearly explained in detail.

---

## Round 0.2 · accepted · Accept

I am happy to inform you that the reviewers are satisfied with the revisions made. Therefore I am provisionally accepting the manuscript for publication.

·

Basic reporting

like i laid out in the last review, the article meets all the criteria . i already gave the submission an "accept"

Experimental design

like i laid out in the last review, the article meets all the criteria . i already gave the submission an "accept"

Validity of the findings

like i laid out in the last review, the article meets all the criteria . i already gave the submission an "accept"

Additional comments

like i laid out in the last review, the article meets all the criteria . i already gave the submission an "accept"

·

Basic reporting

No comment

Experimental design

no comment

Validity of the findings

no comment

Reviewer 3 ·

Basic reporting

Accept.

Experimental design

Accept.

Validity of the findings

Accept.